# Associations between Depressive Symptoms and Satisfaction with Meaningful Activities in Community-Dwelling Japanese Older Adults

**DOI:** 10.3390/jcm9030795

**Published:** 2020-03-14

**Authors:** Michio Maruta, Hyuma Makizako, Yuriko Ikeda, Hironori Miyata, Atsushi Nakamura, Gwanghee Han, Suguru Shimokihara, Keiichiro Tokuda, Takuro Kubozono, Mitsuru Ohishi, Kounosuke Tomori, Takayuki Tabira

**Affiliations:** 1Department of Rehabilitation, Medical Corporation, Sanshukai, Okatsu Hospital, 3-95, Masagohonmachi, Kagoshima 890-0067, Japan; 2Doctoral Program of Clinical Neuropsychiatry, Graduate School of Health Science, Kagoshima University, 8-35-1, Sakuragaoka, Kagoshima 890-8544, Japan; yuriko@health.nop.kagoshima-u.ac.jp (Y.I.); miyata.h01@kumamoto-hsu.ac.jp (H.M.); anakamura@nimd.go.jp (A.N.); hans11057@gmail.com (G.H.); 3Department of Physical Therapy, School of Health Sciences, Faculty of Medicine, Kagoshima University, 8-35-1, Sakuragaoka, Kagoshima 890-8544, Japan; makizako@health.nop.kagoshima-u.ac.jp; 4Department of Occupational Therapy, School of Health Sciences, Faculty of Medicine, Kagoshima University, 8-35-1, Sakuragaoka, Kagoshima 890-8544, Japan; 5Faculty of Health Science, Department of Rehabilitation, Division of Occupational Therapy, Kumamoto Health Science University, 325, Izumimachi, Kita-ku, Kumamoto 861-5598, Japan; 6National Institute for Minamata Disease, Ministry of the Environment, 4058-18 Hama, Minamata, Kumamoto 867-0008, Japan; 7Department of Neuropsychiatry, Kumamoto University Hospital, Kumamoto, Japan, 1-1-1 Honjo Chuo-ku, Kumamoto 860-8556, Japan; 8Master’s Program of Occupational Therapy, School of Health Sciences, Faculty of Medicine, Kagoshima University, 8-35-1, Sakuragaoka, Kagoshima 890-8544, Japan; k5848730@kadai.jp (S.S.); gomyway.k.t@icloud.com (K.T.); 9Department of Cardiovascular Medicine and Hypertension, Graduate School of Medical and Dental Sciences, Kagoshima University, Kagoshima 890-0075, Japan; kubozono@cepp.ne.jp (T.K.); ohishi@m2.kufm.kagoshima-u.ac.jp (M.O.); 10Department of Occupational Therapy, School of Health Science, Tokyo University of Technology, 5-23-22, Nishikamata, Ota-Ku, Tokyo 144-8535, Japan; tomoriks@stf.teu.ac.jp

**Keywords:** meaningful activities, satisfaction, depressive symptoms, community-dwelling older adults, epidemiology

## Abstract

The aim of this cross-sectional study was to investigate relationships between individuals’ ratings of satisfaction and performance of activities that they found meaningful and depressive symptoms. Data was obtained from 806 older adults (mean age 74.9 ± 6.3 years, women = 63.0%) who participated in a community-based health check survey (Tarumizu Study 2018). Participants selected meaningful activities from 95 activities using the Aid for Decision-Making in Occupation Choice and evaluated their satisfaction and performance. Depressive symptoms were assessed using the 15-item Geriatric Depression Scale (GDS-15) and defined by a GDS-15 score of ≥5. Non-linear logistic regression analyses were used separately by gender to examine the association between satisfaction and performance of meaningful activities and depressive symptoms. The prevalence of depressive symptoms was 15.8%. We found no significant difference between meaningful activity choice between older adults with depressive symptoms and those without, in both men and women. After adjusting for potential covariates, satisfaction was associated with depressive symptoms in both men (OR 0.52, 95% CI 0.35–0.77) and women (OR 0.67, 95% CI 0.49–0.91), but performance was limited in women (OR 0.87, 95% CI 0.77–0.99). Our findings suggest that depressive symptoms are associated with satisfaction in meaningful activities regardless of activity categories.

## 1. Introduction

Depressive symptoms and depression are some of the most common mental health problems in older adult populations. Worldwide, there were approximately 322 million people living with depression in 2015 [1], and 8–22% of community-dwelling older adults experienced depressive symptoms [2,3,4]. Depressive symptoms lead to adverse health outcomes such as functional disability [5,6,7], decreased quality of life [8], mortality [6,9], and increased economic costs [10]. The treatment and prevention of depression remain public health priorities [11], and prevention strategies to reduce adverse outcomes associated with depressive symptoms are crucial to these efforts.

Previous studies on community-dwelling older adults have reported associations between depressive symptoms and physical [12,13], cognitive [14], social [15,16], and leisure activities [17,18]. Thus, it is believed that engaging in these activities in daily life can contribute to the prevention or improvement of depressive symptoms. A prospective cohort study investigated behavioral protective factors for depressive symptoms and reported that men who used personal computers and engaged in light physical exercise, and women who participated in community events and attended community meetings experienced protective effects against depressive symptoms [19]. Notably, although various activities have been found to have positive effects on mental health [12,13,14,15,16,17,18,19,20], activities are considered meaningful depending on variable factors, including individual preference, culture, and context. Recently, there has been a growing recognition that engaging in personally valued activities (meaningful activities) are also beneficial for well-being in older adults [21,22]. In 2002, the World Health Organization developed the concept of “Active Aging” as a response to the progress of global aging. Active Aging places considerable value on individuals’ participation in activities that they find meaningful [23]. In order to improve mental health in older adults, not only participation in activities, but also whether the activities are meaningful to an individual can be important. Some meaningful activities (e.g., cooking meals, dressing, and bathing) are routine, while others include work, caring for others, social activities, and leisure activities [24]. Generally, satisfaction with activities is related to depression and subjective well-being. Moreover, satisfaction with activities has been found to promote well-being [25,26]. Therefore, participation in meaningful activities and satisfaction with those activities are important factors for maintaining mental health.

Many existing studies regarding the effects of daily activities on depressive symptoms do not take meaningfulness and satisfaction into account. Everard reported that the reasons for engaging in activities are more important for older adult well-being than the number and frequency of activities [27]. In order to develop more effective strategies for helping older adults improve mental health through activities, it is important to further examine the associations between satisfaction, performance, and meaningfulness of activities and depressive symptoms. Furthermore, the protective effects of activities on depressive symptoms differ by gender [19], and the characteristics of meaningful activities may also vary according to gender. Therefore, the purpose of this study was (1) to investigate what the meaningful activities performed by community-dwelling older adults are according to gender, with and without depressive symptoms; and (2) to clarify the relationship between satisfaction and performance of those activities and depressive symptoms.

## 2. Materials and Methods

### 2.1. Study Population

This cross-sectional study used data from the Tarumizu Study 2018. The Tarumizu Study 2018 was a community-based health check conducted in collaboration with Kagoshima University (Faculty of Medicine), Tarumizu City Office, and Tarumizu Chuo Hospital, from June to December of 2018. In order to be included in the study, participants were required to be residents of Tarumizu City and aged ≥65 years at the time of examination. Exclusion criteria were history of dementia and psychotropic drug use. We also excluded missing data from the meaningful activity assessment, Mini-Cog (cognitive function assessment), walking speed, and grip strength tests. At the end of the exclusion process, data from 806 participants (mean age 74.9 ± 6.3 years, women = 63.0%) were analyzed (Figure 1). This study was approved by the Kagoshima University (Faculty of Medicine) Ethics Committee (Ref No. 170351, approval date: 26 October 2018), and informed consent was obtained from all participants prior to their inclusion in the study.

### 2.2. Measures

#### 2.2.1. Depressive Symptoms

Depressive symptoms were assessed using the 15-item Geriatric Depression Scale (GDS-15) [28], which is suitable for screening depression in community-dwelling older adults [29]. A cutoff point of 4/5 was used to define the presence of depressive symptoms [30].

#### 2.2.2. Meaningful Activities

In the current study, meaningful activity was operationally defined as “the activities that individuals consider important in their current daily life.” The Aid for Decision-Making in Occupation Choice (ADOC), which is an application that can be found on iPad (Apple, Cupertino, CA, USA), was used to collect data regarding meaningful activities [31]. ADOC was developed to identify activities that are meaningful to clients in rehabilitation [31] (an English-language version has also been developed) [32]. The activities in ADOC are depicted by 95 illustrations related to the “activities and participation” included in the International Classification of Human Functioning, Disability, and Health (ICF) and consist of the following components: self-care, mobility, domestic life, work/education, interpersonal interaction, social life, sport, and leisure. During interviews with investigators, participants selected 3 to 5 meaningful activities from the ADOC. Subsequently, the participants rated their satisfaction with each activity using a scale of 1–5 (1: very dissatisfied, 5: very satisfied). The reliability and validity of individualized satisfaction score in ADOC has been reported [33]. Furthermore, participants rated their performance on a scale of 1–10 (1: with great difficulty, 10: perfectly). Well-trained assistant staff investigated meaningful activities. In this study, we examined the activities that participants considered most meaningful.

#### 2.2.3. Covariates

Data on sociodemographic variables including age (years), gender, body mass index (BMI), living status, and education (years) were collected, along with information on medications (n/day) and medical history. The covariates in this study comprised these sociodemographic variables, cognitive function, physical function, and social engagement. Doctors and nurses interviewed participants regarding medical history and medications.

Cognitive function was assessed using the Mini-Cog test, which consisted of a three-word recall task and the clock drawing test [34]. The Mini-Cog detects cognitive impairment more accurately than the Mini-Mental State Examination and is less affected by language and education level [35,36]. The total score is the sum of the correct words recalled (0–3) and the clock drawing (0 or 2) scores, and a cut-off score of <3 best distinguishes between those with cognitive impairment and those without [34,35]. Therefore, in the present study, a total score <3 was defined as poor cognitive status.

Physical function scores were based on maximum grip strength and normal walking speed. The maximum grip strength scores had different cutoff values based on sex (<26 kg for men, <18 kg for women) [37], and walking speed scores had a cutoff value of <1.0 m/s [38]. Participants with either or both scores below the cut-off value were considered to have poor physical status.

Social engagement was assessed using subitems in the Japan Science and Technology Agency Index of Competence [39,40]. Social engagement (i.e., participate in a neighborhood association, participate in regional events, engage in charity, assume a managerial position in a residents’ association) is scored on a dichotomous rating scale (0 = “no”, 1 = “yes”), and the item scores are summed to obtain a total score (range 0–4). Higher scores reflect a higher level of social engagement.

### 2.3. Statistical Analysis

All analyses were conducted using IBM SPSS Statistics version 25.0 (IBM Corp., Armonk, NY). *P* values < 0.05 were considered statistically significant. To compare characteristics between the depressive group and the non-depressive group, we used Pearson’s χ^2^ tests for categorical variables, student’s t-test for continuous variables, and Mann–Whitney U test for ordinal variables. Pearson’s χ^2^ tests were used to compare the proportion of meaningful activity categories by gender and were subjected to residual analysis.

#### 2.3.1. Association between Depressive Symptoms and Meaningful Activity Categories by Gender

The proportion of meaningful activities categories selected by individuals with and without depressive symptoms was calculated by gender, and Fisher’s Exact Test was used to compare them.

#### 2.3.2. Association between Depressive Symptoms, Satisfaction, and Performance

Non-linear logistic regression analyses were used to examine the association between satisfaction, performance, meaningful activity, and depressive symptoms. There were three regression models: the crude model, adjusted model 1, and adjusted model 2. In each model, presence of depressive symptoms was set as a dependent variable. In the crude model, satisfaction and performance were individually set as independent variables. In adjusted model 1, satisfaction and performance were individually set as independent variables and adjusted for potential covariates. In adjusted model 2, satisfaction and performance were simultaneously set as independent variables and adjusted for potential covariates. Potential covariates included age, gender, body mass index, living status, social engagement, education, medications, cognitive function, and physical function. In addition, for the separate analysis on each gender, non-linear logistic regression analyses were also conducted among men and women individually.

## 3. Results

### 3.1. Characteristics of the Participants

Characteristics of the study participants are represented in Table 1. Among the 806 subjects, 127 (15.8%) had depressive symptoms. Participants with depressive symptoms were older (*p* = 0.010), had lower levels of satisfaction (*p* < 0.001) and performance (*p* = 0.005) for meaningful activities, had higher rates of poor physical status (*p* < 0.001), used more medication (*p* = 0.084), and were less socially engaged (*p* < 0.001) than those without depressive symptoms. Of the activity categories, leisure (23.6%) was selected the most often, followed by interpersonal interactions (19.1%), and domestic life (17.9%) (Figure 2). A significantly higher percentage of men chose the leisure (men: 30.9%, women: 19.3%; *p* < 0.01) and work/education categories (men: 10.4%, women: 4.7%; *p* < 0.01), and a higher percentage of women chose domestic life (men: 9.7%, women: 22.6%; *p* < 0.01) (Figure 2).

### 3.2. Association between Depressive Symptoms and Meaningful Activity Categories

Figure 3 illustrates the meaningful activity categories chosen by the depressive and non-depressive groups by gender. For both men (*p* = 0.120) and women (*p* = 0.088), there were no differences in activity choice between the depressive group and the non-depressive group. The men most often selected leisure (non-depressive group: 30.3%, depressive group: 34.0%), followed by interpersonal interactions (non-depressive group: 19.5%, depressive group: 6.4%), sports (non-depressive group: 12.7%, depressive group: 12.8%), and work/education (non-depressive group: 11.2%, depressive group: 6.4%). The women most often selected domestic life (non-depressive group: 21.3%, depressive group: 30.0%), followed by interpersonal interactions (non-depressive group: 19.9%, depressive group: 21.3%), leisure (non-depressive group: 19.2%, depressive group: 20.0%), and self-care (non-depressive group: 12.4%, depressive group: 15.0%). Furthermore, the activities most often selected among the sub-items in each meaningful activity categories are as follows: self-care; “eating/drinking”, “maintaining one’s health”, mobility; “driving”, domestic life; “cooking meals”, “assisting old people/patients”, “child care”, work/education; “remunerative employment”, “non-remunerative employment”, interpersonal interaction; “family relationships”, “friendship”, social life; “religion (e.g., visit a grave)”, sport; “croquet”, “walking”, and leisure; “horticulture” (see Appendix A).

### 3.3. Association between Depressive Symptoms, Satisfaction, and Performance

Table 2 shows the results of the univariate and multivariate logistic regression analyses. The univariate logistic regression analyses showed that both satisfaction (OR 0.60, 95% CI 0.49–0.73, *p* < 0.001) and performance (OR 0.86, 95% CI 0.78–0.93, *p* = 0.001) were significantly related to depressive symptoms (crude model). Even after adjusting for potential covariates, satisfaction (OR 0.61, 95% CI 0.49–0.75, *p* < 0.001) and performance (OR 0.89, 95% CI 0.81–0.98, *p* = 0.019) were significantly related to depressive symptoms (adjusted model 1). The multivariate logistic regression analysis, in which satisfaction and performance were applied simultaneously, showed that only satisfaction (OR 0.62, 95% CI 0.49–0.79, *p* < 0.001) was significantly related to depressive symptoms (adjusted model 2).

Table 3 shows the results of the separate analysis on each gender. Satisfaction with activity was significantly associated with depressive symptoms, even after adjusting for potential covariates, in both men (adjusted model 2: OR 0.52, 95% CI 0.35–0.77, *p* = 0.001) and women (adjusted model 2: OR 0.67, 95% CI 0.49–0.91, *p* = 0.009). In men, activity performance and depressive symptoms were not related in any of the models, whereas in women, performance and depressive symptoms were significantly related even after adjusting for potential covariates (adjusted model 2: OR 0.87, 95% CI 0.77–0.99, *p* = 0.041).

## 4. Discussion

This cross-sectional study showed that meaningful activity categories did not differ significantly between older adults with depressive symptoms and those without. After adjusting for potential covariates, activity satisfaction was associated with depressive symptoms in both men and women, but performance was limited in women. Thus, our results suggest that depressive symptoms are associated with satisfaction in meaningful activities regardless of activity categories.

In this study, men performed leisure and work activities, while women performed domestic life and interpersonal interaction; this is similar to the findings reported in a study that investigated gender differences in social and leisure activities among Japanese older adults [41]. Additionally, Weller and Corey reported that much of the physical activity done by women was household chores [42]. Other studies have reported a relationship between participation in various activities in daily life and depressive symptoms [12,13,14,15,16,17,18,19], but no studies have investigated the activities that community-dwelling older adults with depressive symptoms find meaningful. Depressive symptoms in older adults lead to the limitation of activities, including social activities, in daily life [43,44], and poor life-space mobility [45]. In the current study, males with depressive symptoms tended to choose domestic life and leisure, and fewer tended to choose interpersonal interactions and work; and women with depressive symptoms tended to choose domestic life, while fewer tended to choose sports. In both cases, there were no significant differences in meaningful activity categories for participants with or without depressive symptoms. This suggests that the presence of depressive symptoms does not affect what community-dwelling older people find meaningful. However, this finding should be interpreted with caution, as meaningfulness can be influenced by culture, individual preferences, and context.

The finding that satisfaction is related to depressive symptoms regardless of activity category is a major strength of this study. Previous studies have suggested several mechanisms by which engaging in physical and social activities can positively affect depressive symptoms [46,47]. Reviews regarding the antidepressant mechanisms of physical activity have described biological effects on neuroplasticity, inflammation, oxidative stress, and endocrine systems and psychosocial effects on self-esteem, social support, and self-efficacy, but it is unclear which mechanism has the most sufficient evidence [46]. Social activities have protective effects against depression including the stimulation of multiple bodily systems, development of coping strategies that may reduce the risk of depression, and reinforcement of an individual’s attachment to other psychosocial resources [47]. Meanwhile, meaningful activities give one’s life purpose, fulfill important personal or cultural goals [24,48], and enhance life satisfaction. Several studies of community-dwelling older adults have shown that life satisfaction is related to depression [49,50,51]. Additionally, studies reported that the presence of meaning in life is positively linked to health-related quality of life and perceived health [52,53], and Van der Heyden et al. showed that the presence of meaning in life is associated with low levels of depressive symptoms [54]. Further, it has been suggested that a sense of meaning derived from daily activities will ultimately bring meaning to life [55]. Our findings suggest that engaging in satisfying meaningful activities brings meaning to life, increases life satisfaction, and has a positive effect on depressive symptoms.

In this study, we investigated meaningful activities as “the activities that individuals consider important in their current daily life”. The concept includes various factors such as individual interests and preferences, current life situations and roles, and necessity. It may be important for community-dwelling older adults to evaluate individuals comprehensively and to support meaningful activities in line with their individual values.

Our study had some limitations. First, we assessed depressive symptoms based on screening tools rather than on full diagnostic procedures. Second, since this study is a cross-sectional study design, we cannot confirm the causal relationship between depressive symptoms and satisfaction with meaningful activities. The fact that higher satisfaction with activities is associated with low depressive symptoms may also mean that participants with low levels of depressive symptoms will appreciate activity satisfaction. Additionally, depression is a major risk factor for activity limitations [43,44], and the association between depressive symptoms and meaningful activities may be bidirectional. Longitudinal investigation is necessary to clarify the causal relationship. Third, the data obtained in this study is from a single city. Since meaningful activities in daily life are also affected by geographic characteristics, generalization is limited. Finally, 859 older people (which represented approximately 15% of older people in the city) were enrolled, and they were not randomly selected.

## 5. Conclusions

This study found a significant association between satisfaction with meaningful activities and depressive symptoms in community-dwelling older adults. Supporting activities that are meaningful and give satisfaction to older individuals may contribute to the development of strategies for maintaining mental health.

## Figures and Tables

**Figure 1 jcm-09-00795-f001:**
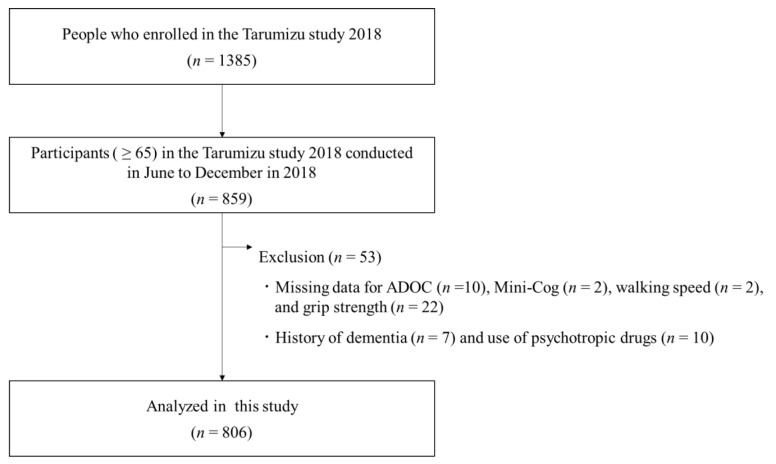
Flowchart of inclusion and exclusion criteria for this study.

**Figure 2 jcm-09-00795-f002:**
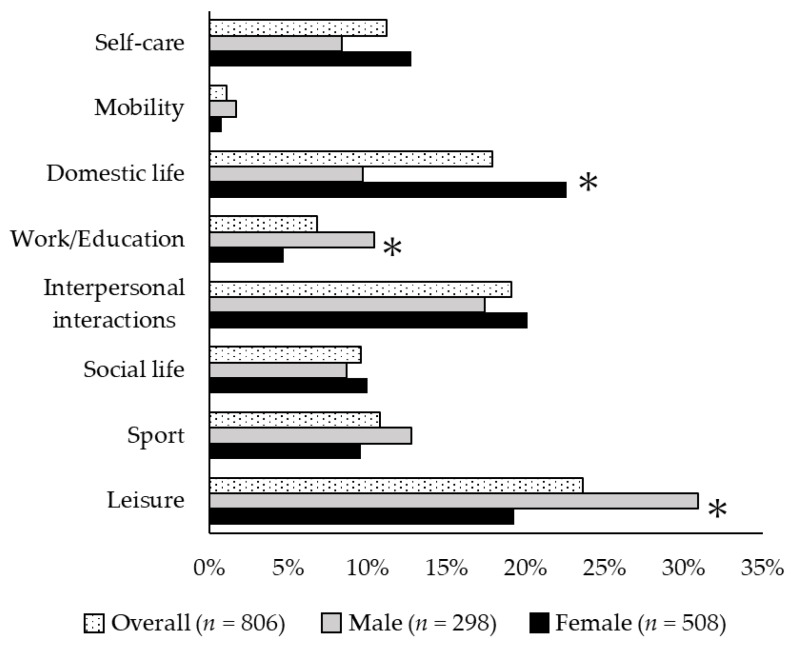
Comparison of activities participants found meaningful. The proportion of activity categories men and women consider to be meaningful is shown. Leisure (men: 30.9%, women: 19.3%; *p* < 0.01) and work/education (men: 10.4%, women: 4.7%; *p* < 0.01) were more frequently chosen by men, while domestic life (men: 9.7%, women: 22.6%; *p* < 0.01) was most frequently chosen by women. Other activities did not differ by gender.

**Figure 3 jcm-09-00795-f003:**
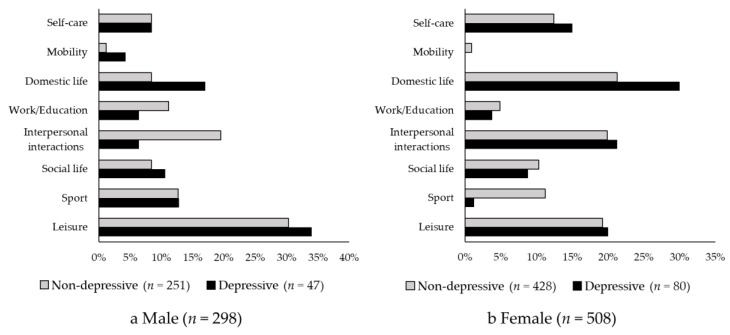
Proportions of meaningful activity categories for non-depressive and depressive groups by gender. (**a**) Proportions of each meaningful activity category in men. (**b**) Proportions of each meaningful activity category in women. There were no significant differences in the proportions of meaningful activity categories depending on the presence or absence of depressive symptoms, in both men (*p* = 0.120) and women (*p* = 0.088).

**Table 1 jcm-09-00795-t001:** Characteristics of the participants.

	All Participants (*n* = 806)	Non-Depressive Group (*n* = 679)	Depressive Group (*n* = 127)	*p* Value
Age, mean ± SD (years)	74.9 ± 6.3	74.7 ± 6.2	76.2 ± 6.5	0.010 ^a^
Female, *n* (%)	508(63.0)	428 (63.0)	80 (63.0)	0.993 ^b^
Satisfaction with Activity, Median (IQR)	5.0 (4.0–5.0)	5.0 (4.0–5.0)	4.0 (3.0–5.0)	<0.001 ^c^
Performance of Activity, Median (IQR)	10.0 (8.0–10.0)	10.0 (8.0–10.0)	8.0 (6.0–10.0)	0.005 ^c^
Poor cognitive Status, *n* (%)	168 (20.8)	137 (20.2)	31 (24.4)	0.281 ^b^
Poor Physical Status, *n* (%)	249 (30.9)	192 (28.3)	57 (44.9)	<0.001 ^b^
Education, Mean ± SD (years)	11.2 ± 2.3	11.2 ± 2.3	10.8 ± 2.2	0.084 ^a^
BMI, Mean ± SD (kg/m^2^)	23.3 ± 3.4	23.3 ± 3.3	23.0 ± 3.6	0.286 ^a^
Medications, Mean ± SD (Number)	4.2 ± 4.6	4.0 ± 4.2	5.3 ± 6.2	0.022 ^a^
Social Engagement, Median (IQR)	3.0 (1.0–4.0)	3.0 (2.0–4.0)	2.0 (0–3.0)	<0.001 ^c^
Living Alone, *n* (%)	211 (26.2)	170 (25.0)	41 (32.3)	0.088 ^b^

SD, standard deviation; IQR, interquartile range; BMI, body mass index; a Student’s t-test, b Pearson’s χ^2^ test, c Mann–Whitney U test.

**Table 2 jcm-09-00795-t002:** Association between depressive symptoms, satisfaction, and performance.

	Crude Model			Adjusted Model 1						Adjusted Model 2		
	OR	95% CI	*p* Value	OR	95% CI	*p* Value	OR	95% CI	*p* Value	OR	95% CI	*p* Value
Satisfaction with Activity	0.60	0.49–0.73	<0.001	0.61	0.49–0.75	<0.001				0.62	0.49–0.79	<0.001
Performance of Activity	0.86	0.78–0.93	0.001				0.89	0.81–0.98	0.019	0.97	0.88–1.08	0.598
Age				1.00	0.96–1.04	0.954	1.00	0.96–1.03	0.795	1.00	0.96–1.04	0.941
Gender				0.85	0.55–1.31	0.457	0.85	0.55–1.30	0.446	0.85	0.55–1.32	0.464
Body Mass Index				0.98	0.92–1.04	0.412	0.97	0.91–1.03	0.294	0.97	0.92–1.04	0.400
Living Alone				1.11	0.69–1.78	0.673	1.22	0.76–1.94	0.411	1.11	0.69–1.78	0.662
Education				0.97	0.88–1.07	0.557	0.97	0.88–1.06	0.488	0.97	0.88–1.07	0.530
Medications				1.02	0.98–1.06	0.332	1.03	0.99–1.07	0.200	1.02	0.98–1.06	0.334
Poor Cognitive Status				1.03	0.61–1.73	0.911	1.00	0.60–1.66	0.998	1.02	0.61–1.72	0.938
Poor Physical Status				1.26	0.78–2.04	0.348	1.21	0.75–1.95	0.442	1.24	0.77–2.02	0.377
Social Engagement				0.62	0.53–0.73	<0.001	0.62	0.53–0.72	<0.001	0.64	0.54–0.73	<0.001

OR, odds ratio; CI, confidence interval; In each model, presence of depressive symptoms was set as a dependent variable; Crude model: Satisfaction and performance were individually set as independent variables; Adjusted model 1: Satisfaction and performance were individually set as independent variables, and adjusted for age, gender, body mass index, living style, social engagement, education, medications, cognitive function, and physical function; Adjusted model 2: Satisfaction and performance were simultaneously set as independent variables, and adjusted for age, gender, body mass index, living style, social engagement, education, medications, cognitive function, and physical function.

**Table 3 jcm-09-00795-t003:** Association between depressive symptoms, satisfaction, and performance in each gender.

	Crude Model			Adjusted Model 1			Adjusted Model 2		
	OR	95% CI	*p* Value	OR	95% CI	*p* Value	OR	95% CI	*p* Value
Men									
Satisfaction with Activity	0.59	0.43–0.80	0.001	0.61	0.44–0.85	0.004	0.52	0.35–0.77	0.001
Performance of Activity	0.97	0.83–0.97	0.649	1.03	0.88–1.21	0.719	1.18	0.88–1.08	0.091
Women									
Satisfaction with Activity	0.60	0.47–0.78	<0.001	0.59	0.44–0.78	<0.001	0.67	0.49–0.91	0.009
Performance of Activity	0.80	0.71–0.89	<0.001	0.82	0.73–0.92	0.001	0.87	0.77–0.99	0.041

OR, odds ratio; CI, confidence interval; In each model, presence of depressive symptoms was set as a dependent variable; Crude model: Satisfaction and performance were individually set as independent variables; Adjusted model 1: Satisfaction and performance were individually set as independent variables, and adjusted for age, body mass index, living style, social engagement, education, medications, cognitive function, and physical function; Adjusted model 2: Satisfaction and performance were simultaneously set as independent variables, and adjusted for age, body mass index, living style, social engagement, education, medications, cognitive function, and physical function.

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
