# Peer review of "Associations between Depressive Symptoms and Satisfaction with Meaningful Activities in Community-Dwelling Japanese Older Adults"

_jcm, 2020, doi:10.3390/jcm9030795_

Round 1

Reviewer 1 Report

This is an interesting study. Though its conclusions do not seem innovative, the manuscript is very well written, the analysis seems overall well performed and the sample size is quite significant. I have some comments and  recommend some revisions:

  1. Abstract:

The cross-sectional design of this study needs to be stated either in the title or in the abstract, as these are the two items potential readers first screen before having access to the manuscript.

The authors state that the association was adjusted for “potential covariates”. It would be good if they could state them in the abstract.

  1. Introduction:

Lines 90-91: The aim “Therefore, the purpose of this study was to investigate meaningful activities of community-dwelling older adults according to gender, and to clarify the relationships between activity satisfaction and performance, and depressive symptoms.” needs to be more clear. It seems there are two aims: (1) to analyze what are the meaningful activities performed by older adults, especially by older women and older men, with and without depressive symptoms; (2) to measure the association of satisfaction and performance while performing these activities with depressive symptoms. The authors should rewrite the aim.

Considering this aim, the results in the abstract should state the differences between gender.

Also considering this aim, the two analyses should be clearly separated on the Methods section, under the statistical analysis subsection. Dividing the text into two paragraphs would be enough. Moreover, for a higher consistency, the authors should consider performing the second part of the study stratified by gender.

  1. Methods:

Lines 98-99: It is not clear why do authors exclude older adults with history of stroke and depression. For the first, they later adjust the analysis for physical and cognitive function which could be the main confounders in this case. For the second, the exclusion of older adults with history of depression may decrease the representativeness the sample and the extrapolation of the results. I would recommend including both groups in the analysis.

Line 100: The authors should state the meaning of Mini-cog as it is only later described.

Line 125: The answer ‘not at all’ to ‘performance’ does not clearly state what is asked. The authors should clarify.

Line 129: The authors should opt either by the word “sex” or “gender” and use it through the entire manuscript.

Covariates subsection: If information on hearing impairment is available it should be considered as a covariate as it is correlated with social isolation and this with depression.

Lines 154-155: It is not clear what the authors mean with “proportion of categories of meaningful activities in the depressive and non-depressive groups”. Is it the frequency of participation of individuals in activities’ (by categories), per sex and depression status?

Line 157: It is not clear how do the authors treat “meaningful activity” in the univariate and multivariate logistic analyses. Was it treated as dependent variable or as a covariate? This should be clarified in the models’ description as well as in Table 2.

  1. Results:

Figure 2 description is insufficient. The figure lists the activities performed and considered meaningful by men and women.

Figures 2 and 3: The authors should use the same colour scheme in both graphs.

Table 2: The authors should present he coefficients (or ORs) for each covariate. As stated previously, the analysis should be performed by gender, as the weight of satisfaction and performance should vary with the meaningful activities usually performed by women (higher frequency of performance of domestic life) could differ from those performed by men (higher frequency of performance of leisure activities).

5. Discussion:

Line 234: “Preferred” may not be the best word as it transmits a ‘free’ willingness to choose these tasks. Though, in most societies women are more frequently impelled to perform domestic tasks, and even the main responsible for those, while men are not. I would modify the word to a more ‘pragmatic’ one (for instance, “perform”).

Lines 270-271: The authors correctly mention the incapacity of the study to measure effect or causation. Though, they should expose into more detail different interpretations of the masured association (indeed, if higher satisfaction with activities can be related with lower risk for depression, the inverse, lower depression related with a higher satisfaction with performed activities, could also be read).

The authors should also discuss better the concept of "meaningful". Considering the Methods’ description, these are “the activities that individuals consider important in their current daily life” This importance can be given because they like to do it, because they were taught through their life they should perform or even excel in these tasks, or because these are fundamental for their survival or autonomy (especially domestic tasks for women). Though, as the authors study older adults in community-dwellings, the importance of the last point (the need to perform these tasks) can be diluted, which gives strength to this study.

6. Conclusion:

Lines 281-282: The sentence is not fully supported by this study design and results (as this is a cross sectional study). I would suggest a more cautious approach: “Supporting activities that are meaningful and give satisfaction to older individuals may contribute to the development of strategies for maintaining mental health.”

Author Response

We appreciate your careful reading of our manuscript and your thoughtful suggestions to improve it. We have made revisions based on your comments and suggestions, as shown below. Your original comments are indicated for reference, and our point-by-point responses are given below.

This is an interesting study. Though its conclusions do not seem innovative, the manuscript is very well written, the analysis seems overall well performed and the sample size is quite significant. I have some comments and recommend some revisions:

  1. Abstract:

Point 1: The cross-sectional design of this study needs to be stated either in the title or in the abstract, as these are the two items potential readers first screen before having access to the manuscript.

Response 1: Thank you for pointing this out. We have revised this accordingly (p. 1, line 34).

Point 2: The authors state that the association was adjusted for “potential covariates”. It would be good if they could state them in the abstract.

Response 2: We appreciate your comment. However, because we added content to the abstract about the gender analysis, we did not have space to specify what the “potential covariates” were.

  1. Introduction:

Point 3: Lines 90-91: The aim “Therefore, the purpose of this study was to investigate meaningful activities of community-dwelling older adults according to gender, and to clarify the relationships between activity satisfaction and performance, and depressive symptoms.” needs to be more clear. It seems there are two aims: (1) to analyze what are the meaningful activities performed by older adults, especially by older women and older men, with and without depressive symptoms; (2) to measure the association of satisfaction and performance while performing these activities with depressive symptoms. The authors should rewrite the aim.

Response 3: We appreciate your comment. As you suggested, we revised our sentence because the previous version obscured the purpose of the study (p. 2, lines 87-90).

Point 4: Considering this aim, the results in the abstract should state the differences between gender.

Response 4: We appreciate your comment. The results have been described based on gender in line with the revision of the purpose of this study (p. 1, lines 44-47).

Point 5: Also considering this aim, the two analyses should be clearly separated on the Methods section, under the statistical analysis subsection. Dividing the text into two paragraphs would be enough. Moreover, for a higher consistency, the authors should consider performing the second part of the study stratified by gender.

Response 5: We thank you for pointing this important information out. We divided the statistical analysis into two subsections, in line with the revised objectives. In addition, we conducted separate analysis on each gender (p. 4, lines 156-170).

  1. Methods:

Point 6: Lines 98-99: It is not clear why do authors exclude older adults with history of stroke and depression. For the first, they later adjust the analysis for physical and cognitive function which could be the main confounders in this case. For the second, the exclusion of older adults with history of depression may decrease the representativeness the sample and the extrapolation of the results. I would recommend including both groups in the analysis.

Response 6: We thank you for pointing this important information out. As you noted, the representativeness of the sample and the extrapolation of the results may be decreased. We have revised the exclusion criteria and performed an analysis that included older adults with a history of stroke and depression (p 3., lines 97-98, Figure 1).

Point 7: Line 100: The authors should state the meaning of Mini-cog as it is only later described.

Response 7: Thank you for pointing this out. We have added the meaning of Mini-cog (p. 3, line 99).

Point 8: Line 125: The answer ‘not at all’ to ‘performance’ does not clearly state what is asked. The authors should clarify.

Response 8: Thank you for pointing this out. We have revised this to “with great difficulty” to clarify the answer to “performance” (p. 3, line 125).

Point 9: Line 129: The authors should opt either by the word “sex” or “gender” and use it through the entire manuscript.

Response 9: Thank you for pointing this out. We reviewed the entire manuscript and unified the expression, opting to use “gender”.

Point 10: Covariates subsection: If information on hearing impairment is available it should be considered as a covariate as it is correlated with social isolation and this with depression.

Response 10: We appreciate your comment. We also consider hearing impairment to be important information. However, information on hearing impairment was not included in the covariates because it was obtained from only a small number of participants.

Point 11: Lines 154-155: It is not clear what the authors mean with “proportion of categories of meaningful activities in the depressive and non-depressive groups”. Is it the frequency of participation of individuals in activities’ (by categories), per sex and depression status?

Response 11: We appreciate your comment. We mean the proportion of activity categories selected by individuals with and without depressive symptoms. We have revised this accordingly (p. 4, lines 157-158).

Point 12: Line 157: It is not clear how do the authors treat “meaningful activity” in the univariate and multivariate logistic analyses. Was it treated as dependent variable or as a covariate? This should be clarified in the models’ description as well as in Table 2.

Response 12: Thank you for pointing this out. Meaningful activities were set as independent variables, and the presence of depressive symptoms was set as a dependent variable. We have added this point to the text (p. 4, lines 162-163) and revised Table 2 as well.

  1. Results:

Point 13: Figure 2 description is insufficient. The figure lists the activities performed and considered meaningful by men and women.

Response 13: Thank you for pointing this out. We have added to the description of Figure 2 (p. 7, lines 185-189).

Point 14: Figures 2 and 3: The authors should use the same colour scheme in both graphs.

Response 14: Thank you for this comment. The color scheme is different between Figures 2 and 3 because the content to be expressed is different.

Point 15: Table 2: The authors should present he coefficients (or ORs) for each covariate. As stated previously, the analysis should be performed by gender, as the weight of satisfaction and performance should vary with the meaningful activities usually performed by women (higher frequency of performance of domestic life) could differ from those performed by men (higher frequency of performance of leisure activities).

Response 15: We appreciate your comment. We have added the OR for each covariate (Table 2). In addition, we have added the results of a separate logistic regression analysis on each gender (p. 8, lines 222-228, Table 3).

  1. Discussion:

Point 16: Line 234: “Preferred” may not be the best word as it transmits a ‘free’ willingness to choose these tasks. Though, in most societies women are more frequently impelled to perform domestic tasks, and even the main responsible for those, while men are not. I would modify the word to a more ‘pragmatic’ one (for instance, “perform”).

Response 16: We agree with you. We have revised this accordingly (p. 11, line 241).

Point 17: Lines 270-271: The authors correctly mention the incapacity of the study to measure effect or causation. Though, they should expose into more detail different interpretations of the masured association (indeed, if higher satisfaction with activities can be related with lower risk for depression, the inverse, lower depression related with a higher satisfaction with performed activities, could also be read).

Response 17: We appreciate your comment. As you point out, participants with lower levels of depressive symptoms may value satisfaction with activities. We have added this point to the limits of the study (p.11-12, lines 283-285).

Point 18: The authors should also discuss better the concept of "meaningful". Considering the Methods’ description, these are “the activities that individuals consider important in their current daily life” This importance can be given because they like to do it, because they were taught through their life they should perform or even excel in these tasks, or because these are fundamental for their survival or autonomy (especially domestic tasks for women). Though, as the authors study older adults in community-dwellings, the importance of the last point (the need to perform these tasks) can be diluted, which gives strength to this study.

Response 18: We thank you for pointing this important information out. We also consider the concept of meaningful activities to be very important. As you pointed out, it is necessary to describe the support for community-dwelling older adults that can be given from the concept of meaningful activity in this study. We have added this point (p. 11, lines 275-279).

  1. Conclusion:

Point 19: Lines 281-282: The sentence is not fully supported by this study design and results (as this is a cross sectional study). I would suggest a more cautious approach: “Supporting activities that are meaningful and give satisfaction to older individuals may contribute to the development of strategies for maintaining mental health.”

Response 19: We agree with you. We have revised this accordingly (p. 12, lines 293-295).

Reviewer 2 Report

This study examines the association between depression and satisfaction with meaningful activities among 760 Japanese elderly adults. The study findings are meaningful and provide important research and practical implications.

A few minor issues could be addressed to improve the quality of the study.

  1. The use of linear regression model on the numeric dependent variable makes sense. But depression score is usually heavily skewed. Were there diagnostic tests conducted to examine any violations to OLS regression assumptions?
  2. The authors included the satisfaction as a numeric variable in the regression, when in fat the variable is an ordinal/factor variable and should be treated as so in the regression model. Did the author try including satisfaction as a factor variable in the regression models?
  3. Are there any differences in demographic characteristics and depression scores between study sample and the 99 adults who were excluded?
  4. Did the authors run regression separately for men and women? 
  5. The authors have any speculations on how different types of meaningful activities might affect depression differently? Might be worth mentioning in the Discussion section.

Author Response

We appreciate your careful reading of our manuscript and your thoughtful suggestions to improve it. We have made revisions based on your comments and suggestions, as shown below. Your original comments are indicated for reference, and our point-by-point responses are given below.

This study examines the association between depression and satisfaction with meaningful activities among 760 Japanese elderly adults. The study findings are meaningful and provide important research and practical implications.

A few minor issues could be addressed to improve the quality of the study.

Point 1: The use of linear regression model on the numeric dependent variable makes sense. But depression score is usually heavily skewed. Were there diagnostic tests conducted to examine any violations to OLS regression assumptions?

Response 1: We appreciate your comment. We used non-linear logistic regression analyses and the dependent variable was the presence or absence of depressive symptoms. We have corrected the expression in the manuscript (p. 1, line 40, p. 4, lines 160, 169).

Point 2: The authors included the satisfaction as a numeric variable in the regression, when in fat the variable is an ordinal/factor variable and should be treated as so in the regression model. Did the author try including satisfaction as a factor variable in the regression models?

Response 2: Thank you for pointing this out. We included satisfaction with activities and activity performance with activities as ordinal variables in the regression models.

Point 3: Are there any differences in demographic characteristics and depression scores between study sample and the 99 adults who were excluded?

Response 3: Thank you for pointing this out. We have revised our exclusion criteria. Participants with dementia and a history of psychotropic drug use were excluded. Therefore, the 53 older adults who were excluded had a higher GDS score (p = 0.036) and used more medication (p = 0.042). There were no differences in other demographic characteristics.

Point 4: Did the authors run regression separately for men and women?

Response 4: We thank you for pointing this important information out. As you noted, separate regression analysis by gender has important implications for this study. We have added the separate analysis on each gender (p. 4, lines 169-170, p. 8, lines 222-228).

Point 5: The authors have any speculations on how different types of meaningful activities might affect depression differently? Might be worth mentioning in the Discussion section.

Response 5: We appreciate your comment. We believe that the satisfaction with activities that older people find to be meaningful affects depression regardless of the types of activities; the text reflects this (p. 11, lines 239-240, 257-258).